# Transcriptomic and Proteomic Analyses Unveil the Role of Nitrogen Metabolism in the Formation of Chinese Cabbage Petiole Spot

**DOI:** 10.3390/ijms25031366

**Published:** 2024-01-23

**Authors:** Ying Mei, Juanli Lei, Wenqi Liu, Zhichen Yue, Qizan Hu, Peng Tao, Biyuan Li, Yanting Zhao

**Affiliations:** 1Institute of Vegetables, Zhejiang Academy of Agricultural Sciences, Hangzhou 310021, Chinataopeng-84@163.com (P.T.); 2College of Horticulture Science, Zhejiang A&F University, Hangzhou 311300, China

**Keywords:** petiole spot, nitrogen metabolism, glutamine synthetase, glutamate dehydrogenase, reactive oxygen species, cell death

## Abstract

Chinese cabbage is the most widely consumed vegetable crop due to its high nutritional value and rock-bottom price. Notably, the presence of the physiological disease petiole spot significantly impacts the appearance quality and marketability of Chinese cabbage. It is well known that excessive nitrogen fertilizer is a crucial factor in the occurrence of petiole spots; however, the mechanism by which excessive nitrogen triggers the formation of petiole spots is not yet clear. In this study, we found that petiole spots initially gather in the intercellular or extracellular regions, then gradually extend into intracellular regions, and finally affect adjacent cells, accompanied by cell death. Transcriptomic and proteomic as well as physiology analyses revealed that the genes/proteins involved in nitrogen metabolism exhibited different expression patterns in resistant and susceptible Chinese cabbage lines. The resistant Chinese cabbage line has high assimilation ability of NH_4_^+^, whereas the susceptible one accumulates excessive NH_4_^+^, thus inducing a burst of reactive oxygen species (ROS). These results introduce a novel perspective to the investigation of petiole spot induced by the nitrogen metabolism pathway, offering a theoretical foundation for the development of resistant strains in the control of petiole spot.

## 1. Introduction

Chinese cabbage (*Brassica rapa* L. ssp. *Pekinensis*), originating in China, is the most common and frequently consumed vegetable crop in East Asia due to its high nutritional content including vitamins, minerals, and glucosinolates, as well as its rock-bottom price [1,2,3]. Chinese cabbage can be supplied all year round due to its strong adaptability and storage ability, which are illustrated by the fact that Chinese cabbage can be produced in spring, summer, and autumn, and in addition, it can be stored for 3–4 months at low temperature. A large number of diseases, including parasitic and nonparasitic ones, may occur during the production and storage processes. These affect the yield and quality of Chinese cabbage. Parasitic diseases are brought about by the attack of bacteria, fungi, or a virus. For example, soft rot disease, one of the three most economically important diseases of Chinese cabbage, is caused by the pathogen *Erwinia carotovora* ssp. *carotovora* (*Ecc*) [4]. However, there are many disease conditions in which no parasite is involved, termed as nonparasitic diseases or disorders, which result from the detrimental influence of certain climatic and soil conditions, physiological imbalances, or genetic abnormalities. Outstanding examples of nonparasitic diseases are soil mineral deficiency disorders, flooding injury, and freezing and heat damage, leading to symptoms such as wilting, discoloration, and an overall decline in Chinese cabbage health [4,5].

Notably, a novel physiological disease, namely, petiole spot, has occurred among Chinese cabbage in the past four decades. The phenotype comprises numerous sesame-like spots, which are black or blackish brown in color, primarily concentrated in the epidermal cells of the petiole. The size of petiole spot typically ranges from 1 to 2 mm. As the condition worsens, the area and density increase, along with the petiole epidermis cracking or rotting [6,7]. Petiole spot has become a serious threat to the quality of the appearance and the marketability of Chinese cabbage, causing significant losses.

The disease’s occurrence results from a complex interplay between genotype and environmental factors [8]. A fertilizer application experiment showed that elevated nitrogen levels exacerbate petiole spot, leading to an increased density of spots [9]. In addition, the occurrence of petiole spot in Chinese cabbage is also linked to high copper (Cu) levels and low boron (B) levels [10]. Systemic observation in the field found that pH value is also an important influencing factor for disease occurrence. Soil with a pH of 8.0 is particularly suited for the onset of this disease [6]. Other factors such as air temperature, low light levels, soil nutrient levels, and postharvest methods also contribute to development of petiole spot [11]. Significantly, excessive application of nitrogen fertilizer is recognized as a crucial factor, and different forms of nitrogen have various effects, with ammonium nitrogen having a stronger impact than nitrate and amide nitrogen [12]. It is noteworthy that previous research on Chinese cabbage petiole spot mainly focuses on field investigations and physiological experiments and has clarified the influencing factors contributing to the onset of the disease. However, the precise mechanism underlying Chinese cabbage petiole spot remains unclear, and more experimental evidence is needed to confirm this. Although the promotion of Chinese cabbage petiole spot by excessive nitrogen fertilizer is well established, the precise mechanism remains elusive.

To unravel the intricate mechanisms of Chinese cabbage petiole spot, we meticulously examined the morphology of petiole spot in varieties that are resistant and susceptible to petiole spot. Transcriptome and proteome analyses were employed to identify differentially expressed genes (DEGs) and differentially expressed proteins (DEPs) among resistant and susceptible varieties upon high nitrogen treatment. The findings of our study could provide new insights into breeding strategies for resistance to petiole spot in Chinese cabbage.

## 2. Results

### 2.1. Phenotypic Investigation of Resistance and Susceptibility to Petiole Spot in Chinese Cabbage

Two Chinese cabbage varieties, Han Chunwa C8-1 and 4-2-3-4, known for their extreme characteristics of petiole spots, were employed as research materials. The variety of Han Chunwa C8-1(R) is deemed highly resistant due to few spots being observed on the petiole. Meanwhile, the variety of Han Chunwa 4-2-3-4 (S) exhibited a susceptible phenotype, which has many spots of inconsistent size on the petiole (Figure 1A). Six spots with different sizes (I-VI) were selected from the susceptible variety for microscope observation. The findings revealed that the spots are indeed brown and initially gather in the intercellular or extracellular regions. Subsequently, they gradually extend into intracellular areas, and finally, they affect multiple cells as the size of the spots increases (Figure 1B).

Additionally, cells without petiole spots (CK) and those with petiole spots (petiole spot) on Chinese cabbage were examined using an electron microscope, and it was found that petiole spot showed significant cellular changes. These alterations involve the distortion of the cell wall, the presence of multiple vesicles, and the accumulation of black particulate matter, as compared to that which is observed in normal cells (Figure 2A). These distinctive features seem to correspond with the typical characteristics related to cell death in plants. Trypan blue staining was used to validate the hypothesis. The result showed that the cells within the petiole spot as well as their neighboring cells were observed to be blue in color, whereas the normal cells were not stained, verifying that cell death occurred within and around the areas with spots (Figure 2B).

### 2.2. Transcriptome and Proteome Profiling of Resistant and Susceptible Chinese Cabbage Lines upon Nitrogen Treatment

Previous research has indicated that petiole spot on Chinese cabbage worsened with high levels of nitrogen treatment. To investigate the mechanisms of the high nitrogen levels involved in petiole spot, we conducted transcriptomic and proteomic analyses on resistant and susceptible Chinese cabbage lines upon nitrogen treatment. Samples treated with ddH_2_O for the resistant variety are named as CKA, and ddH_2_O-treated samples for the susceptible variety are named as CKB. Similarly, samples treated with 10 mM NH_4_NO_3_ for the resistant variety are named as TA, and those for the susceptible variety are named as TB. A total of 83.06 Gb of clean data were obtained from 12 transcriptome samples, with each sample’s clean data amounting to 6.00 Gb. The percentage of Q30 bases was 94.05% or higher. FPKM values were utilized to standardize the expression in each sample and examine the gene expression pattern further. Pairwise comparisons of each treatment group revealed 177, 150, 1485, and 1234 up-regulated genes and 236, 95, 1023, and 1264 down-regulated genes in the CKA vs. TA, CKB vs. TB, CKA vs. CKB, and TA vs. TB comparisons, respectively (Figure 3A). Most of the differentially expressed genes were identified in CKA vs. CKB and TA vs. TB, with only a small number observed in CKA vs. TA and CKB vs. TB comparisons (Figure 3B).

Additionally, proteomic assays were conducted to illuminate the mechanisms associated with elevated nitrogen levels contributing to the development of petiole spot. A total of 120,752 peptides were identified, with 91,781 being specific peptides. Specifically, 7718 quantifiable proteins were detected. Setting a *p*-value of <0.05 and employing a differential expression change of more than 1.3 as the threshold for significant up-regulation and less than 1/1.3 as the threshold for significant down-regulation, 3292, 584, and 361 up-regulated proteins were identified in each comparative group, CKA vs. TA, CKB vs. TB, CKA vs. CKB, and TA vs. TB, respectively (Figure 3C). We also identified 10,158, 696, and 342 down-regulated proteins. Aligned with the transcriptome, the majority of the differentially expressed proteins were identified in CKA vs. CKB and TA vs. TB, with only a handful being discovered in CKA vs. TA and CKB vs. TB comparisons (Figure 3D).

### 2.3. Integrated Proteomic and Transcriptomic Analysis

A comprehensive analysis was conducted on transcriptomics and proteomics data. Out of all the detected genes, 9842 were identified on the protein map (Figure 4A). The correlation analysis showed a correlation coefficient of 66.2% between quantified proteins and their corresponding mRNAs (Figure 4B). Gene Set Enrichment Analysis (GSEA) was utilized to identify biological pathways associated with different phenotypes through the enrichment of predefined gene/protein sets within a ranked list of genes. The results revealed that 17 pathways shared the same regulatory relationship between the transcriptome and proteome, including pathways of photosynthesis, linoleic acid metabolism, pentose and glucuronate interconversion, porphyrin and chlorophyll metabolism, alpha-linolenic acid metabolism, alanine, aspartate and glutamate metabolism, tropane, piperidine and pyridine alkaloid biosynthesis, peroxisome, arginine biosynthesis, glyoxylate and dicarboxylate metabolism, beta-alanine metabolism, carbon fixation in photosynthetic organisms, nitrogen metabolism, glycolysis/gluconeogenesis, carbon metabolism, and metabolic pathways and biosynthesis of secondary metabolites(Figure 4C).

In the results of grouping and sorting by hierarchical clustering analysis, genes and proteins identified in transcriptome and proteome analysis were classified into six clusters with distinct expression patterns (Figure 4D). Notably, the nitrogen metabolism pathway (map00910) was enriched both in cluster1 and cluster 6 (Figure 4E,F). In summary, genes involved in nitrogen metabolism exhibited different expression patterns in resistant and susceptible Chinese cabbage lines, indicating that the distinct phenotype of petiole spot in resistant and susceptible Chinese cabbage lines might result from the different N metabolism ability.

### 2.4. Nitrogen Metabolism Pathway in Resistant and Susceptible Chinese Cabbage Lines

Considering that nitrogen metabolism might affect the development of petiole spots, we screened the differentially expressed genes and identified 11 genes that were strongly correlated with the nitrogen metabolism (Pearson correlation coefficients > 0.8, *p*-values < 0.05) at the transcriptomic or proteomic level. These genes encompass one nitrate transporter (NTR) gene (Brassica_rapa_newGene_963), three nitrate reductase (NR) genes (BraA07g026900.3c, BraA02g024390.3c, BraA07g040960.3c), two nitrite reductase genes (BraA07g005240.3c, BraA09g010360.3c), three glutamine synthetase (GS) genes (BraA10g023190.3C, BraA04g010460.3C, BraA02g006550.3), one glutamate dehydrogenease (GDH) gene (BraA01g043400.3C), and one NADH-dependent glutamate synthase (GOGAT) gene (BraA03g014440.3C).

According to gene function analysis, the data of gene expression showed that the expression of Brassica_rapa_newGene_963 (NTR), BraA10g023190.3C, BraA04g010460.3C, BraA02g006550.3C (GS), and BraA01g043400.3C (GDH) was higher in CKA compared to those in CKB. Furthermore, expression levels of those genes were lower TB compared to CKB. Conversely, the expression levels of BraA07g026900.3c, BraA02g024390.3c, and BraA07g040960.3c (NR) were lower in Chinese cabbage that was resistant to petiole spot when compared to the sensitive cabbage. Additionally, their expression was lower in sensitive Chinese cabbage treated with high nitrogen as compared to the control. Notably, both GS and GDH are important enzymes in nitrogen metabolism, as they have a crucial function in converting inorganic nitrogen to organic nitrogen, thus reducing the toxicity of NH_4_^+^. In general, the resistant Chinese cabbage line has high assimilation ability of NH_4_^+^, highlighting the importance of GS and GDH in the development of petiole spot in Chinese cabbage.

To further corroborate our hypothesis, we assessed the ammonium (NH_4_^+^-N) levels as well as enzyme activities of GS and GDH in Chinese cabbage varieties that exhibit resistance and susceptibility to petiole spot upon high nitrogen treatments. We found a notably increased level of NH_4_^+^ in the susceptible Chinese cabbage compared to the resistant variety (Figure 5B). Moreover, the distinction between the resistant and susceptible lines was magnified when there was exposure to high levels of nitrogen. The enzyme activities of GS and GDH in the susceptible sample (B) were notably low compared to those in the resistant sample. The enzyme activities of NH_4_^+^ glutamine synthetase and glutamate dehydrogenase in the susceptible material experienced a slight increase following high nitrogen treatment (Figure 5C,D). The results demonstrate that low activities of GS and GDH are responsible for the accumulation of NH_4_^+^ in the susceptible line. These were exacerbated by high levels of nitrogen fertilizer, thereby intensifying the occurrence of petiole spots.

### 2.5. Expression Analysis of Genes Related to ROS Production and Scavenging

The staining results suggest that the occurrence of petiole spot is correlated with cell death (Figure 2). As a burst of reactive oxygen species (ROS) consistently coincides with cell death, we analyzed the expression of genes involved in ROS production and scavenging. Notably, 53 differentially expressed genes strongly correlated with ROS production and scavenging were identified (Pearson correlation coefficients > 0.8, *p*-values < 0.05) at the transcriptomic or proteomic level (Figure 6).

Among the identified DEGs, 10 genes were contributors to ROS production, including BraA10g005020.3C, BraA09g043070.3C (acyl-CoA oxidase, ACX), BraA02g009250.3C (aldehyde oxidase, AAO), BraA09g016610.3C (superoxide dismutase, SOD (Cu-Zn)), BraA09g006630.3C (SOD (Fe)), and BraA01g039910.3C (SOD (Mn)). Compared to the resistant counterpart, the susceptible line showed higher expression levels in BraA01g021740.3C, BraA08g008660.3C, BraA09g032560.3C (ferredoxin, Fd), and BraA09g012740.

As for ROS scavenging, forty-three genes were identified, sixteen of which exhibited increased expression in CKB compared to those in CKA, whereas the remaining genes displayed the opposite pattern of expression. The same trend is present in TB compared to CKB. The development of petiole spot in Chinese cabbage appears to be intricately linked to ROS production in the leaves. The delicate balance between ROS production and scavenging emerges as a crucial factor in differentiating susceptibility from resistance to petiole spot.

### 2.6. ROS Accumulation in Petiole Cells of Resistant and Susceptible Chinese Cabbage Lines

H_2_DCFDA fluorescence staining was utilized on the epidermis of Chinese cabbage leaves that exhibited petiole spot. Significant elevations in green fluorescence were observed in cells within the petiole spot and their adjacent areas compared to normal cells, indicating an increase in ROS content within the petiole spot cells (Figure 7A).

This indicates a need for further investigation into differentially expressed genes. To investigate ROS levels in the epidermal cells of petiole spot-resistant and -susceptible cabbages under normal and high nitrogen conditions, our study found an increased accumulation of ROS in susceptible cabbage in comparison to the resistant variety. This accumulation was intensified further under high nitrogen treatment, with green fluorescence primarily concentrated in the plasmalemma region (Figure 7B).

## 3. Discussion

Petiole spot, as one of the most destructive physiological diseases, wreaks havoc on the petiole, severely impacting the marketability of the affected produce. Lesions manifest as discrete dark brown or black spots, measuring 1 to 2 mm in diameter, on the white petiole. Interestingly, according to our observation in the field, no petiole spots were observed on the rosette leaves of Chinese cabbage. However, the number of petiole spots gradually decreased from the outermost to the inner leaf layers of the head. In the present study, we employed a microscope to illustrate the progression of lesions from small to large over different periods. The findings revealed that the lesions originate in the intercellular or extracellular regions and progressively extend to neighboring cells (Figure 1). Prior research has emphasized the extracellular region and the extracellular body space as primary battlegrounds for host plant and pathogen interactions, where reactive oxygen species (ROS) levels are tightly regulated by plant redox systems [13,14]. Notably, no pathogenic bacteria were identified in Chinese cabbage petiole spot. Nevertheless, the current understanding of the main components of the dark brown material produced by petiole black spot remains incomplete, serving as a stimulus for future research endeavors. In addition, exhibit distinct characteristics, including evident cell wall distortion and deformation, the presence of multiple vesicles, and the deposition of black material. This cellular response indicates a severe state of cell death (Figure 2). Considering that the stratum corneum covering the cell wall acts as a protective barrier against organisms [15], fractures of the external layer result in the exposure of internal cells, which make it easy for pathogenic microorganisms to enter, leading to leaf disease and eventually decay [16]. Cell wall distortion emerges as a potential primary cause of epidermal division in the late stage of Chinese cabbage leaf stalk macula. Through petiole spot may not pose a direct threat to the overall growth of Chinese cabbage, cell wall distortion, deformation, and even death may heighten vulnerability to other pathogens, such as *Erwinia carotovora* ssp. *carotovora* (*Ecc*), causing significant production issues in Chinese cabbage. Therefore, it is necessary to explore the mechanism of petiole spot, in order to effectively control the occurrence of petiole spot and other, even more serious diseases in Chinese cabbage.

It has been well documented in several reports that the excessive application of nitrogen fertilizer can exacerbate the occurrence of petiole spot [9]. In contrast to the abiotic stress affecting Chinese cabbage, plant nutrition plays a crucial role in growth and development. Nutrients play a crucial role in plant growth and development, and deficiencies can compromise plant immunity, significantly impacting overall growth [17,18]. For instance, phosphorus (P) enhances photosynthesis, carbohydrate synthesis, and transport. A deficiency in phosphorus results in blocked carbohydrate metabolism, delayed cell division, and hindered cell elongation [19]. Similarly, copper (Cu), an essential nutrient, serves as a crucial component of many plant enzymes and participates in photosynthetic electron transport. Copper deficiency is characterized by slow growth, a lack of green young leaves, and the appearance of dead spots [20]. However, excessive mineral nutrients can disrupt metabolic processes in plant cells, triggering REDOX reactions and causing cell damage. For example, an excess of phosphate fertilizer strengthens plant respiration, consuming significant amounts of sugar and energy, thereby inhibiting growth [19]. Excess Cu binding to S, N, and O ligands inhibits enzyme and functional metalloprotein activity. Additionally, as a REDOX active metal, Cu+ can be oxidized to form highly toxic reactive oxygen species [21,22,23]. In the same way, nitrogen is indispensable for plant growth and reproduction. It serves as a vital component of proteins and nucleic acids and plays a crucial role in carbon fixation during photosynthesis as a part of chlorophyll [24]. Plants possess the ability to obtain nitrogen either from the atmosphere as nitrogen (N_2_) or by absorbing soil nitrogen through their roots. The primary uptake of nitrogen by the vegetable root system occurs from inorganic nitrogen sources, including ammonium and nitrate, as well as small organic nitrogen-containing compounds present in the soil. Research has demonstrated the pivotal role of nitrogen in promoting rapid plant growth and enhancing yield. Consequently, traditional agricultural practices have often relied extensively on the application of excessive nitrogen fertilizers, aiming to maximize vegetable yields [25,26]. However, a high nitrogen fertilization rate increased the severity of disease caused by botrytis cinerea in lettuce [27]. In Chinese cabbage, excessive nitrogen can induce petiole spot disease [9]. This dual nature of nutrient effects underscores the delicate balance needed for optimal plant health and productivity.

In this study, the analysis of transcriptome and proteome data from resistant and susceptible Chinese cabbage varieties subjected to different nitrogen treatments unveiled the significant involvement of nitrogen metabolic pathways in the process (Figure 3 and Figure 4). The plant nitrogen metabolic pathway facilitates the transport of inorganic nitrogen compounds into leaves through ammonium transporters (AMTs), nitrate transporters (NRT), and amino acid transporters (AATs). Within the plant, nitrate reductase (NR) orchestrates the cytoplasmic reduction of NO_3_^−^ to nitrite (NO_2_^−^), and nitrite reductase (NiR) converts nitrite (NO_2_^−^) to NH_4_^+^ in the plastids. The potential toxic accumulation of ammonium nitrogen poses a threat to plant health, underscoring the crucial role of ammonia assimilation in plants. The process of plant ammonia assimilation typically transforms inorganic nitrogen into organic nitrogen through the GS-GOGAT cycle, initiated by glutamine synthetase (GS) and glutamine-2-oxo-glutamate transaminase/glutamate synthetase (GOGAT) [24,28]. Ammonia assimilation may also occur via glutamate dehydrogenase (GDH) in the presence of high ammonia concentrations. Significant differences in gene expression and enzyme activity of GS and GDH were observed between susceptible and resistant varieties in this study. Additionally, the accumulation of ammonium nitrogen (NH_4_^+^ −N) in sensitive varieties treated with high nitrogen was significantly up-regulated (Figure 5). The emphasis on the importance of ammonia assimilation in the nitrogen metabolism process underscores its role in the onset of petiole spot. It is noteworthy that elevated nitrogen levels exacerbate petiole spot, leading to an increased density of spots, with ammonium nitrogen having a stronger impact than nitrate and amide nitrogen [12]. These findings align with previous research, reinforcing the influence of the nitrogen metabolism pathway on the development of leaf stalk disease in Chinese cabbage.

A differential gene analysis of cabbage resistant and susceptible to petiole spot under normal and high nitrogen treatments revealed several genes related to plant ROS production that were enriched (Figure 6). Cells with petiole spots on Chinese cabbage exhibit a higher abundance of ROS compared to cells without petiole spots (Figure 7A). ROS are generated in response to a variety of environmental stresses, causing oxidative damage that may lead to cell death [29]. Moreover, an increase in ROS levels was observed in susceptible cabbages upon staining for ROS, and high nitrogen treatments exacerbated this discrepancy (Figure 7B). Notably, NH_4_^+^ has been recognized as a catalyst for elevated levels of H_2_O_2_ and sequent oxidative stress responses in various plant species, including Arabidopsis [30], the aquatic plant *Vallisneria natans* [31], and numerous other plant species [32]. Interestingly, ROS were concentrated in the intercellular or extracellular regions, mirroring the initial onset site of petiole spot disease. This finding aligns with phenotypic observations (Figure 2). ROS production happens within the apoplast in stress conditions and is accelerated by the activity of NADPH oxidase (NOX) [33]. The results of a previous study show that under conditions of high nitrogen concentration, susceptibility traits exhibit heightened cell membrane permeability and plasma membrane peroxidation reactions, as evidenced by increased conductivity and elevated levels of MDA (lipid peroxidation products) [34]. Additionally, the over-application of nitrogen fertilizer can lead to an accumulation of ammonium nitrogen in cabbage petioles, potentially causing damage to the normal cell membrane structure [35], which causes the release of polyphenols and follows reaction catalyed by polyphenol oxidase. This is consistent with previous research results, indicating that Chinese cabbage petiole spot can induce a large number of ROS dependent on excessive accumulation of NH_4_^+^.

## 4. Materials and Methods

### 4.1. Plant Materials

Two varieties of Chinese cabbage, namely, Han Chunwa C8-1 (marked with R), exhibiting high resistance, and Han Chunwa 4-2-3-4 (marked with S), displaying susceptibility, were chosen for the investigation of Chinese cabbage petiole spot. Both varieties were developed by the Vegetable Institute of Zhejiang Academy of Agricultural Sciences, Hangzhou, China. Seeds of these varieties were sown in 1/2 Murashige and Skoog medium (MS) and placed in a greenhouse with 14 h of light (25 °C) and 10 h of darkness (16 °C) for 15 days. Subsequently, seedlings were transplanted to floating plates for hydroponic growth over a period of 1 month, utilizing Hoagland’s nutrient solution (Farwood Bio, Shanghai, China). The nutrient solution was then replaced with either ddH_2_O or a 10 mM NH_4_NO_3_ solution for 8 days, three biological replicates per treatment.

Samples collected for transcriptome and proteome analysis were labeled as follows: ddH_2_O-treated Han Chunwa C8-1 samples: CKA; ddH_2_O-treated Han Chunwa 4-2-3-4 samples: CKB; 10 mM NH_4_NO_3_-treated Han Chunwa C8-1 samples: TA; 10 mM NH_4_NO_3_-treated Han Chunwa 4-2-3-4 samples: TB.

### 4.2. RNA Sequencing and Differentially Expressed Gene Analysis

Total RNA was extracted from leaves with the Spin Column Plant Total RNA Purification Kit (Sangon Biotech, Shanghai, China) according to the manufacturer’s instructions, and the quality of RNA was measured by a spectrophotometer (Thermo Fisher Scientific, Waltham, MA, USA). After this, we constructed a double-stranded cDNA Library using the EzyNGS DNA Library Construction Kit (Sangon Biotech, Shanghai, China) according to this protocol, and the libraries were sequenced on an Illumina Sequencing Platform in Biomarker Technologies (Shunyi, Beijing, China).

Raw data were processed to obtain clean data using Trimmomatic (Version 3.90) [36]. This involved removing low-quality reads, including those with more than 10% reads containing N and more than 50% reads with a base quality score (Q) of ≤10 across the entire read. The clean data were mapped to the Brara chliifu V 3.0 reference genome using HISAT2 software [37]. The resulting reads were assembled using String Tie software (StringTie (www.jhu.edu), accessed on 9 January 2023) [38].

The statistically significant differentially expressed genes (DEGs) were identified using DEseq [39] with a threshold adjust *p*-value < 0.05 and |log2FC| ≥ 1. The Kyoto Encyclopedia of Genes and Genomes (KEGG) (https://www.kegg.jp/, accessed on 9 January 2023) database was used for functional annotation of DEGs (https://geneontology.org/, accessed on 9 January 2023). The Tbtools package Heatmap was used to create expression heatmaps.

### 4.3. Protein Extraction and Proteome Analysis

The sample was first ground by liquid nitrogen and then combined with lysis buffer (including 1% TritonX-100, 10 mM dithiothreitol, and 1% Protease Inhibitor Cocktail, 50 μM PR-619, 3 μM TSA, 50 mM NAM, and 2 mM EDTA). An equivalent volume of Tris-saturated phenol (pH 8.0) was added. After centrifugation (4 °C, 10 min, 5000× *g*), the upper phenol phase was transferred to at least four volumes of ammonium sulfate-saturated methanol and incubated at −20 °C for at least 6 h. After centrifugation at 4 °C for 10 min, the remaining precipitate was washed with ice-cold methanol once and ice-cold acetone three times. The protein was redissolved in 8 M urea, and the protein concentration was determined with a BCA kit according to the manufacturer’s instructions. For the quantitative proteomic study of total protein, TMT labeling in Chinese cabbage leaf samples was conducted by PTM Biotechnology Co., Ltd. (Hangzhou, China).

### 4.4. Transcriptome and Proteome Association Analysis

The transcriptome and proteome data of Chinese cabbage were jointly analyzed with the assistance of PTM Biotechnology Co., Ltd. (Hangzhou, China). The mRNA information derived from the transcriptome was integrated with the protein information identified by the proteome to elucidate the corresponding relationships. The *p*-value and difference multiple of the genes and proteins identified through this joint analysis were compared and analyzed using theorem, theorem repeatability, and Pearson correlation analysis. Furthermore, KEGG pathway-based GSEA analysis, expression cluster analysis, and GO and KEGG enrichment analysis were performed to gain insights into the functional implications of the identified genes and proteins.

### 4.5. Determination of Ammonium Nitrogen Content, Glutamine Synthetase and Glutamate Dehydrogenase Activities in Plants

The content of ammonium nitrogen (NH_4_^+^) in plants was determined using a Plant Ammonium Nitrogen Content Assay Kit (Boxbio, Beijing, China) according to the manufacturer’s instructions. Weigh 0.1 g of Chinese cabbage petiole and combine it with 1 mL of the extraction solution. Homogenize the mixture at room temperature, then centrifuge at 10,000× *g* at 4 °C for 10 min, using the supernatant as the test sample. Preheat the spectrophotometer for at least 30 min, set the wavelength to 630 nm, and zero with distilled water. In a 1.5 mL centrifuge tube, add 20 μL of the sample or extract (blank control), along with 490 μL of Reagent I and 490 μL of Reagent II. Thoroughly mix the components and allow the color reaction to occur at 25 °C for 20 min. Transfer 1 mL of the reaction solution to a 1 mL glass cuvette. Measure the absorbance at 630 nm, record the value as A, and calculate ∆A = A (measurement) − A(blank).

Glutamine synthetase (GS) activities were measured according to the instructions from Solarbio. Weigh 0.1 g of Chinese cabbage petiole and add 1 mL of the extraction solution. Homogenize the mixture in an ice bath and centrifuge at 8000× *g* for 10 min at 4 °C. Collect the supernatant and keep it on ice for subsequent measurements. Preheat the zymograph for 30 min, adjusting the wavelength to 540 nm and zeroing it with distilled water. In the measurement tube, combine 160 μL of Reagent I, 70 μL of Reagent III, and 70 μL of the sample. For the control tube, mix 160 μL of Reagent II, 70 μL of Reagent III, and 70 μL of the sample. After thorough mixing and incubating in a 25 °C water bath, add 100 μL of Reagent 4 to both the measurement and control tubes. Allow them to stand for 10 min, then centrifuge the samples at 5000× *g* for 10 min at room temperature. Transfer 200 μL of supernatant from the sample into a 96-well UV plate and determine the absorbance at 540 nm, recording it as B. Utilize the formula ΔB = B (measurement) − B (control) to calculate the difference in absorbance between the assay and control samples.

Glutamate dehydrogenase (GDH) activities were also measured according to the instructions from Solarbio. Weigh approximately 0.1 g of tissue and combine it with 1 mL of extraction solution for ice bath homogenization. Centrifuge at 8000× *g* and 4 °C for 10 min, then carefully collect the supernatant, keeping it on ice for subsequent measurements. Preheat the zymograph for at least 30 min, adjusting the wavelength to 340 nm and zeroing it with distilled water. In a 96-well UV plate, add 10 μL of the sample and 190 μL of the working solution. Ensure thorough mixing, and immediately record the absorbance value C1 at 340 nm for 20 s, followed by the absorbance value C2 after 5 min and 20 s. Calculate ΔC = C1 − C2.

### 4.6. Electron Microscopy

The samples were cut into small pieces (1 × 3 mm sizes) and fixed with 2.5% (*v*/*v*) glutaraldehyde and 2% (*v*/*v*) osmic acid. After washing three times with 0.1 M phosphate buffer, a series of ethanol gradients (50%, 70%, 80%, 90%, 95%, and 100% in distilled, deionized water (*v*/*v*)) and absolute acetone were used to dehydrate. For embedding, the dehydrated samples were immersed in a series of Spurr resin gradients (50% and 75% in acetone (*v*/*v*)) and absolute Spurr. These samples were placed at the bottom of the tube and then polymerized at 70 °C for 24 h. Thin sections were subsequently cut and observed under a transmission electron microscope (8100, ZEISS) [40].

### 4.7. ROS and Trypan Blue Staining

For detection of the intracellular ROS level, ROS-sensitive probe H_2_DCFDA (MCE, Princeton, NJ, USA) was used [41]. Peel the epidermis of Chinese cabbage leaves using tweezers, and immerse them in an MES-KCl buffer (MES 10 mM, KCl 5 mM, CaCl_2_ 50 mM, pH 6.15) for 30 min. Subsequently, perform dark staining with 50 mM H_2_DCFDA for 2 h. After rinsing three times with PBS (MCE, Princeton, NJ, USA), observe the ROS staining using a fluorescence microscope (Leica DM2500 LED, Wetzlar, Germany) with GFP fluorescence.

To assess cell membrane integrity and cell viability, we employed trypan blue stain (Solarbio, Beijing, China). Chinese cabbage leaves’ epidermis was carefully peeled using tweezers and immersed in the trypan blue stain for a minimum of 1 h. Subsequently, the stained sample was placed in a 95% ethanol solution for overnight dehydration. Following complete decolorization, the sample was equilibrated with sterile water for 30 min. Dead cells were then visualized as blue-stained when observed under a fluorescence microscope.

### 4.8. Statistical Analysis

Data were expressed as the mean ± standard error (SE) of three replicates. One-way ANOVA analysis with subsequent Tukey’s test were analyzed for data, and differences were defined as significant at *p* < 0.05. The drawing was created using GraphPad Prism 8.0.

## 5. Conclusions

Transcriptomic and proteomic analyses were utilized in this study to examine differences in genes and proteins in Chinese cabbage varieties displaying resistance and susceptibility to petiole spot, particularly under normal and high levels of nitrogen. The results demonstrated that excessive nitrogen application impacted the expression of genes and proteins related to nitrogen metabolic pathways, specifically glutamine synthetase and glutamate dehydrogenase. Under normal and high nitrogen treatments, the expression of genes and enzyme activity of these proteins were lower in sensitive varieties compared to resistant ones. This resulted in the accumulation of NH_4_^+^ in Chinese cabbage susceptible to petiole spot, which causes ammonia toxicity to plants, hindering its conversion into nutrients that are essential for plant growth. Various genes involved in generating and removing ROS were identified as differentially expressed in the varieties that are resistant and susceptible to petiole spot in Chinese cabbage grown under normal and high nitrogen conditions. Petiole spot significantly increased the ROS content in affected cells compared to normal cells. In Chinese cabbage strains susceptible to petiole spot, low activity of GS and GDH was observed, exacerbated by excessive nitrogen fertilizer. This led to an accumulation of NH_4_^+^ within plant cells, triggering ROS production and ultimately resulting in cell death and the occurrence of petiole spots.

## Figures and Tables

**Figure 1 ijms-25-01366-f001:**
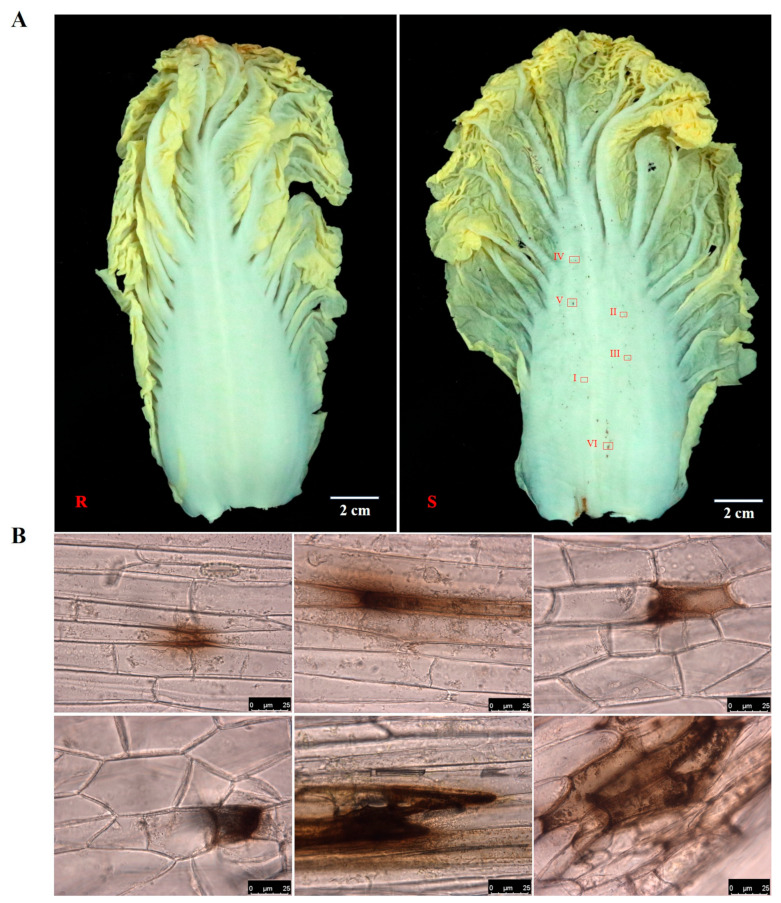
Characteristic of petiole spots. (**A**) Phenotype of petiole spot in Han ChunwaC8-1 (R) and Han Chunwa 4-2-3-4 (S). Han Chunwa C8-1 is resistant to petiole spots, and Han Chunwa 4-2-3-4 (S) is susceptible to petiole spots. (**B**) Microscope image of petiole spots of various sizes on Chinese cabbage (I–VI).

**Figure 2 ijms-25-01366-f002:**
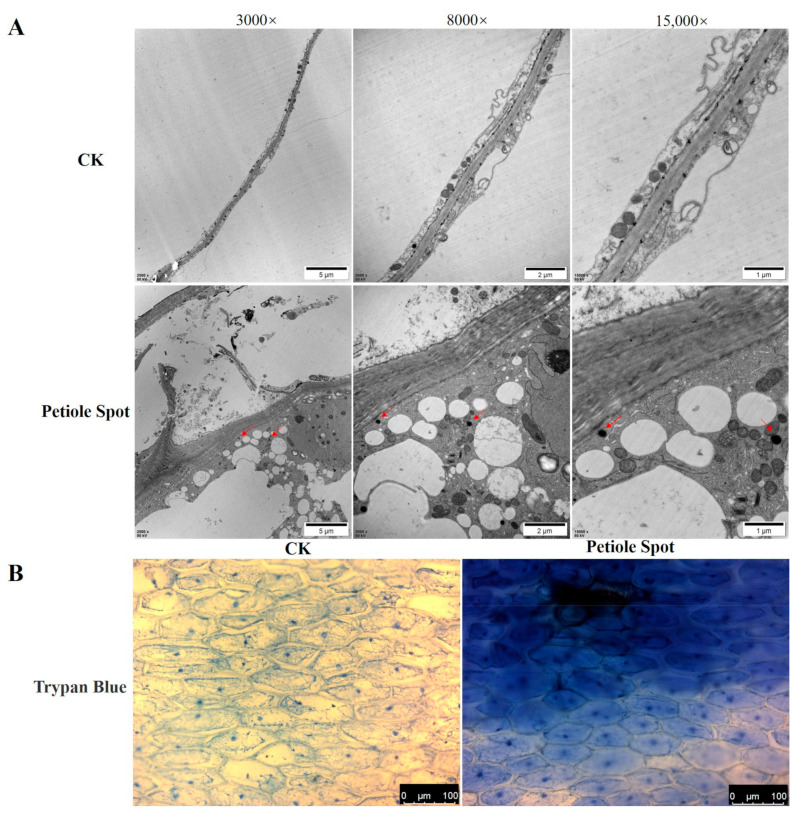
Characteristics of cells within petiole spot areas on Chinese cabbage. (**A**) Cells without petiole spots (CK) and those with petiole spots (petiole spot) on Chinese cabbage were examined using an electron microscope across various magnifications (3000×, 8000×, and 15,000×). The presence of potential black deposits was highlighted by red arrows. (**B**) Trypan blue dye was applied to visualize and stain cells in the vicinity of both those cells without petiole spots (CK) and those with petiole spot areas (petiole spot). Dead cells exhibit a blue coloration.

**Figure 3 ijms-25-01366-f003:**
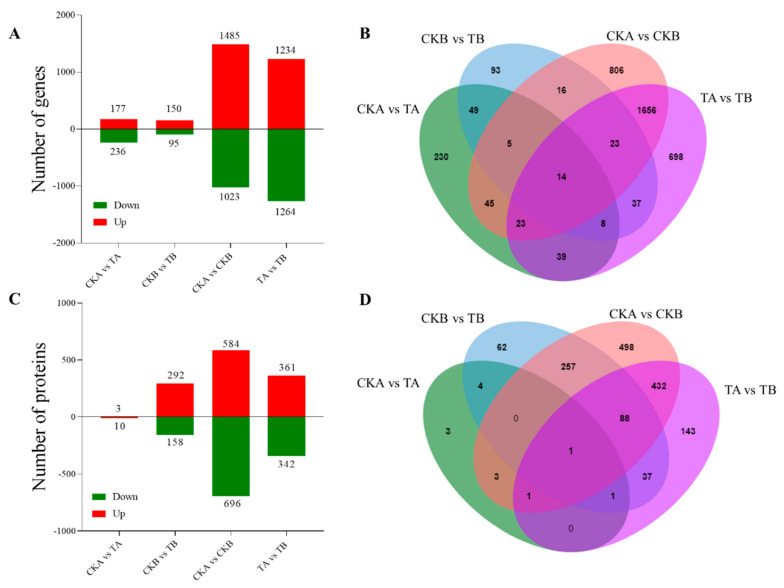
Number of different expression genes and proteins. Different expression genes (DEGs) (**A**) and different expression proteins (DEPs) (**C**) in CKA vs. TA, CKB vs. TB, CKA vs. CKB, and TA vs. TB comparative groups. Venn analysis of DEGs (**B**) and DEPs (**D**) in comparative groups. CKA, ddH_2_O-treated resistant samples. CKB, ddH_2_O-treated susceptible samples. TA, 10 Mm NH_4_NO_3_-treated resistant samples. TB, 10 Mm NH_4_NO_3_-treated susceptible samples.

**Figure 4 ijms-25-01366-f004:**
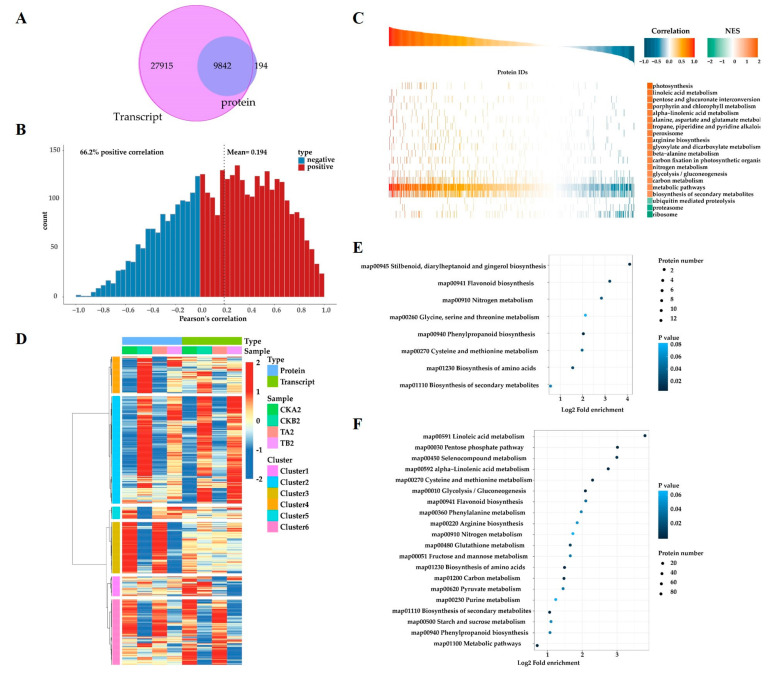
Combined transcriptome and proteome analysis. (**A**) Venn analysis of quantitative comparison of transcriptome and proteome. (**B**) Cumulative distribution of Pearson correlation coefficient between transcriptome and proteome. (**C**) KEGG pathway GSEA analysis based on quantitative correlation coefficient between transcriptome and proteome. (**D**) Transcriptome and proteome expression clustering heat map. (**E**,**F**) KEGG pathway enrichment bubble map of Cluster1 and Cluster6.

**Figure 5 ijms-25-01366-f005:**
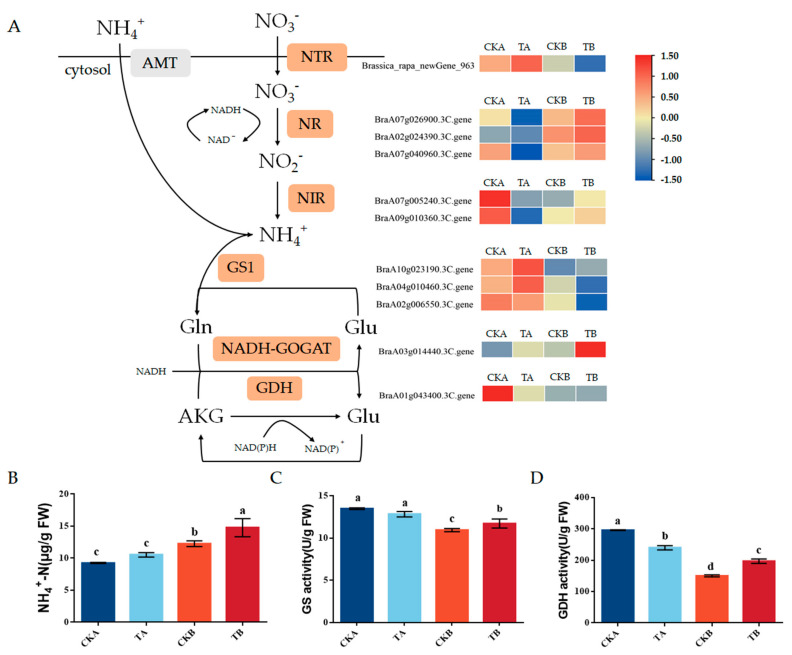
Analysis of expression of genes related to nitrogen metabolism. (**A**) Heat map of expression of nitrogen metabolism pathway and related enzymes. (**B**) The content of ammonium NH_4_^+^-N in different samples. (**C**,**D**) Activity of GS and GDH enzymes in different samples. Values are expressed as the mean ± SE (n = 3). Different letters denote statistically significant differences among the different groups (one-way ANOVA *p* < 0.05, Tukey’s test).

**Figure 6 ijms-25-01366-f006:**
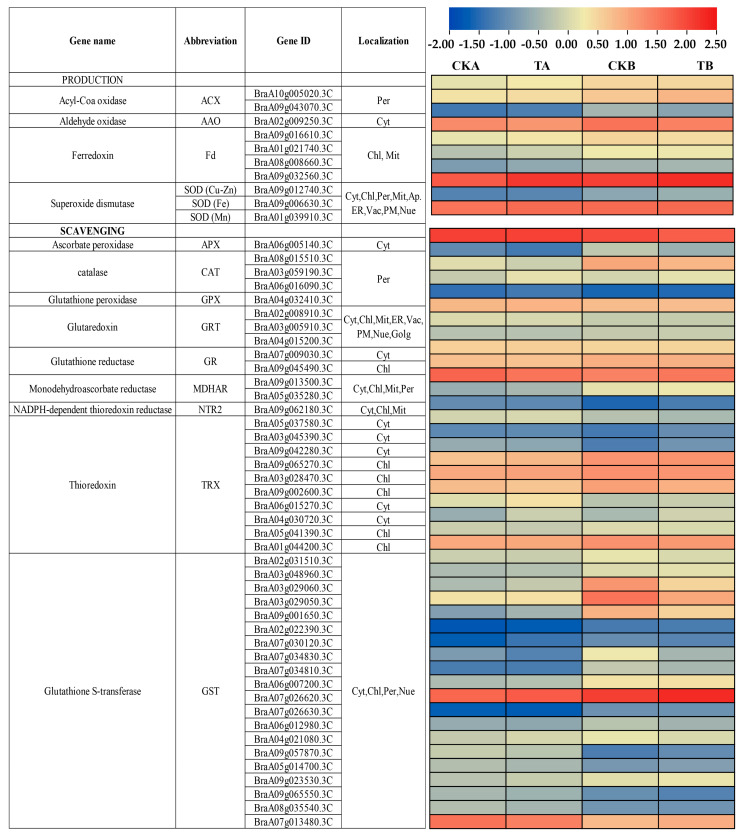
Expression heat maps of ROS regulatory pathway-related genes in different samples.

**Figure 7 ijms-25-01366-f007:**
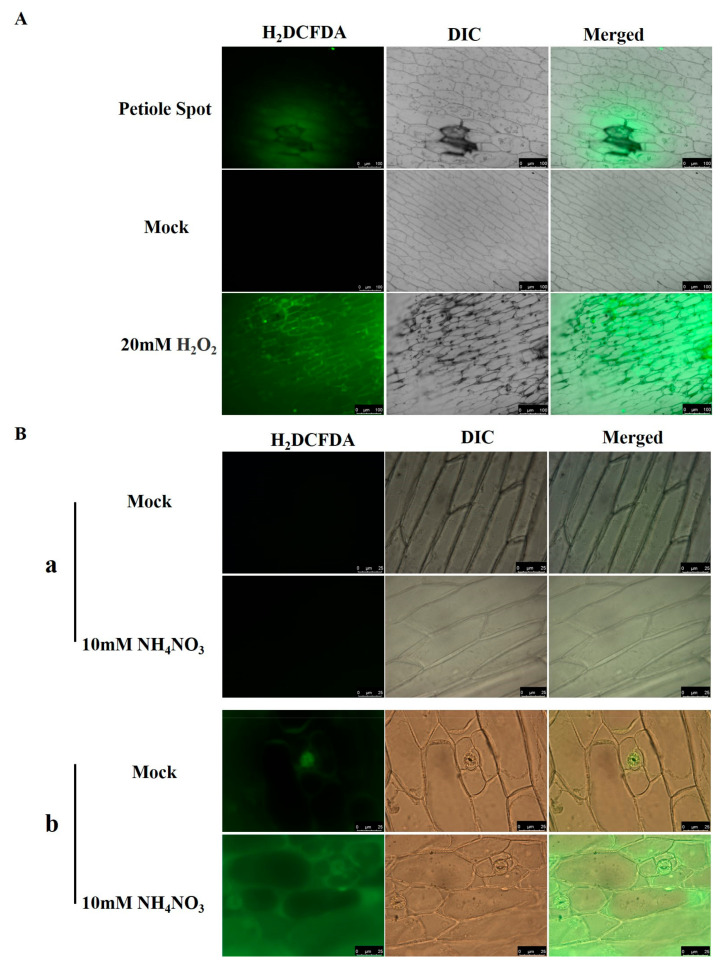
ROS was induced by Chinese cabbage petiole spot. (**A**) The epidermis of Chinese cabbage was stained with H_2_DCFDA, and the control group was stained with either ddH_2_O (mock) or 20 mM H_2_O_2_. (**B**) ROS production in resistant variety (**a**) and sensitive variety (**b**) was observed after 10 mM NH_4_NO_3_ or ddH_2_O (mock) treatment.

## Data Availability

Data are contained within the article.

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
