# Peer review of "Transcriptomic and Proteomic Analyses Unveil the Role of Nitrogen Metabolism in the Formation of Chinese Cabbage Petiole Spot"

_ijms, 2024, doi:10.3390/ijms25031366_

Round 1
Reviewer 1 Report
Comments and Suggestions for Authors
Introduction
Line 30-32. Unclear sentence, needs rephrasing.
It is necessary to clearly formulate the main aim and tasks of the research. It is not clear in the current presentation - whether you are already reporting the results before the work is done. The information given at the end of the Introduction appears to be conclusions.
Literature sources do not need to be presented in the superscript style. This applies to the whole article.
Results
Line 80. “… investigation in the field” - Has it been done before? Then it should be mentioned that it has been clarified in previous studies. As far as I understand from the methodology, the study was conducted in a greenhouse.
It is necessary to make sure that all abbreviations are explained when they are mentioned for the first time in the text. This was especially true of the experimental variants. The reader must clearly understand about which variant of the experiment is given information, discussed. This also applies to Figures. For example, Figure 2 – what is CK? Figures should be completely understandable without going back to the text.
It should be noted that it is very difficult to see scale data in microscope images.
Line 118-119. “… the CKA vs. TA, CKB vs. TB, CKA vs. CKB, and TA vs. TB comparisons…” Since the methodology is given at the end of the article and is also not precise and clear, how can the reader understand what it is about. We can only guess.
Figure 5 should provide some information about the significant differences between treatments. For example, means with different letters were significantly different (p<0.05).
Discussion
In the second paragraph, when talking about the general role of nitrogen in the plant, you should also provide some more fundamental literature sources on plant mineral nutrition, not just a few studies. I would recommend referring to authors such as Marschner, or Mengel and Kirkby or Fageria.
In the final part of the discussion, there is a general lack of any literature references and comparison of the obtained data with the results of research by other authors. There is a feeling that the discussion section was written in a hurry and was not really finished.
If in the Abstract the authors also paid attention to the importance of the work and possible directions of future research, then in the Discussion and Conclusions it is not mentioned at all.
I would recommend improving the Discussions section, focusing more on the relationship of the data obtained in your study with other studies, expanding the range of sources of the literature used.
Materials and methods
More precise information about plant material and experiment variants is required in section 5.1.
“One month after transplantation, the seedlings were continuously cultivated with a 10mM NH4NO3 solution for 8 days and the control groups were treated with sterile water.” But how were the plants cultivated this month? How many plants were used in the study, were there any replications, how was the experiment designed?
“The samples were named CKA, CKB, TA and TB, respectively.” This is an extremely important point in the article. It must be clearly and unambiguously indicated how the control is denoted, how the N variants are denoted.
No information is given about the statistical processing of the data.
References
“3. H, C.; W, K.; T, T.; K, R.; K, Z.; H, W.; T, L. Identification of Key Gene Networks Controlling Soluble Sugar and Organic Acid Metabolism During Oriental Melon Fruit Development by Integrated Analysis of Metabolic and Transcriptomic Analyses. Frontiers in plant science 2022, 13, 830517, doi:10.3389/fpls.2022.830517.” Why are the authors written in this way?
General conclusions
The article creates dual feelings, on the one hand it is interesting, because experimental research always gives new knowledge, contains extensive data material, as well as it is dedicated to such a topical topic as vegetable quality, on the other hand there are many inaccuracies in the article. Suggestions and recommendations for improving the quality of the article are provided in the comments to the authors. I recommend accepting this article in Int. J. Mol. Sci. after minor, but careful revision.
Author Response
Response to Reviewer 1
1、Introduction
Line 30-32. Unclear sentence, needs rephrasing.
Reply:
Thanks for the nice suggestions. We have revised the expression as follow. “Chinese cabbage could be supplied all year around due to its strong adaptability and storage-ability, which are illustrated by the fact that Chinese cabbage could produce in spring, summer and autumn, in addition could be stored for 3-4 month at low temperature.”(Line 30-33)
- It is necessary to clearly formulate the main aim and tasks of the research. It is not clear in the current presentation - whether you are already reporting the results before the work is done. The information given at the end of the Introduction appears to be conclusions. Literature sources do not need to be presented in the superscript style. This applies to the whole article.
Reply:
Thanks for the suggestion. We have clearly formulated the main aim and tasks of the research in the revised manuscript as bellow.
“It is noteworthy that previous research on Chinese cabbage petiole spot mainly focuses on field investigations and physiological experiments and has clarified the influencing factors influencing factors contributing to the onset of the disease. However, the precise mechanism underlying Chinese cabbage petiole spot, remains unclear, which needs more experimental evidence to confirm. Although the promotion of Chinese cabbage petiole spot by excessive nitrogen fertilizer is well-established, the precise mechanism remains elusive.To unravel the intricate mechanisms of Chinese cabbage petiole spot, we meticlously examine the morphology of petiole spot in resistant and susceptible varieties to petiole spot. we Transcriptome and proteome were employed to identify differentially expressed genes (DEGs) and differentially expressed proteins (DEPs) among resistant and susceptible varieties upon high nitrogen treatment. The findings of our study could provide new insights into the breeding strategies for resistance to petiole spot in Chinese cabbage.”(Line 64-76)
- Literature sources do not need to be presented in the superscript style. This applies to the whole article.
Reply:
Thanks for the suggestion. We have updated the source document format for the entire article.
- Results Line 80. “… investigation in the field” - Has it been done before? Then it should be mentioned that it has been clarified in previous studies. As far as I understand from the methodology, the study was conducted in a greenhouse.
Reply:
Thanks for the critical comments. In our previous statement, the description of two Chinese cabbages varieties was not precise. In fact, these varieties are extreme materials identified during our breeding work before the present study. The statement has been revised accordingly as follow.
“Two Chinese cabbage varieties, Han Chunwa C8-1 and 4-2-3-4, known for their extreme characteristics of petiole spots, were employed as research materials.”(Line 80-81)
- It is necessary to make sure that all abbreviations are explained when they are mentioned for the first time in the text. This was especially true of the experimental variants. The reader must clearly understand about which variant of the experiment is given information, discussed. This also applies to Figures. For example, Figure 2 – what is CK? Figures should be completely understandable without going back to the text. It should be noted that it is very difficult to see scale data in microscope images. Line 118-119. “… the CKA vs. TA, CKB vs. TB, CKA vs. CKB, and TA vs. TB comparisons…” Since the methodology is given at the end of the article and is also not precise and clear, how can the reader understand what it is about. We can only guess.
Reply:
We are truly grateful to the critical and thoughtful suggestions. We have added the explanation of all abbreviations appeared in the manuscript. We have provided detailed descriptions of ‘CK’ and ‘Petiole spot’ appeared in FIgure 2 in text (line 90-91) as well as in the legend(line106-107 ). Similarly, detailed descriptions of abbreviation CKA\TA\CKB\TB has been added in text and legend as below. ” Samples treated with ddH2O for the resistant variety are named as CKA, while ddH2O-treated samples for the susceptible variety are named as CKB. Similarly, samples treated with 10 mM NH4NO3 for the resistant variety are named as TA, and those for the susceptible variety are named as TB.”(line 117-121)
“CKA, ddH2O-treated resistant samples. CKB, ddH2O-treated susceptible samples. TA, 10 Mm NH4NO3-treated resistant samples. TB, 10 Mm NH4NO3-treated susceptible samples.” (line 146-148)
Scale data in microscope images has been modified in the revised Figure2. In the new manuscripts, you will find the (Figure 3 and Figure 6).
- Figure 5 should provide some information about the significant differences between treatments. For example, means with different letters were significantly different (p<0.05).
Reply:
Thank you for your suggestions. We have recognized the issue and made the following modifications in the legend of Figure 5 as follow.” Values are expressed as the mean ± SE (n=3). Different letters denote statistically significant differences among the different group (one-way ANOVA P < 0.05, Tukey’s test).” (line 222-224)
- In the second paragraph, when talking about the general role of nitrogen in the plant, you should also provide some more fundamental literature sources on plant mineral nutrition, not just a few studies. I would recommend referring to authors such as Marschner, or Mengel and Kirkby or Fageria.
Reply:
Thank you for your advice. We refer to the articles of several experts as suggestion, and we add the discussion of plant mineral elements in the discussion section as follow.
“Nutrients play a crucial role in plant growth and development, and deficiencies can compromise plant immunity, significantly impacting overall growth. For instance, phosphorus (P) enhances photosynthesis, carbohydrate synthesis, and transport. A deficiency in phosphorus results in blocked carbohydrate metabolism, delayed cell division, and hindered cell elongation . Similarly, copper (Cu), an essential nutrient, serves as a crucial component of many plant enzymes and participates in photosynthetic electron transport. Copper deficiency is characterized by slow growth, a lack of green young leaves, and the appearance of dead spots . However, excessive mineral nutrients can disrupt metabolic processes in plant cells, triggering REDOX reactions and causing cell damage. For example, an excess of phosphate fertilizer strengthens plant respiration, consuming significant amounts of sugar and energy, thereby inhibiting growth. Excess Cu binding to S, N, and O ligands inhibits enzyme and functional metalloprotein activity. Additionally, as a REDOX active metal, Cu+ can be oxidized to form highly toxic reactive oxygen species.” (line 294-307)
- In the final part of the discussion, there is a general lack of any literature references and comparison of the obtained data with the results of research by other authors. There is a feeling that the discussion section was written in a hurry and was not really finished. If in the Abstract the authors also paid attention to the importance of the work and possible directions of future research, then in the Discussion and Conclusions it is not mentioned at all. I would recommend improving the Discussions section, focusing more on the relationship of the data obtained in your study with other studies, expanding the range of sources of the literature used.
Reply:
Thank you for your suggestions. We are so sorry that we didn't organize the discussion section properly, which caused you to have such a misunderstanding. According to your suggestions, we have improved the discussion section totally.
Firstly, we discuss the importance of our study as follow. “ Considering that the stratum corneum covering the cell wall acts as a protective barrier against organisms, the fractures of external layer results in exposure of internal cells, which made it simple for pathogenic microorganisms to enter, leading to leaf disease and decay eventually. Cell wall distortion emerges as a potential primary cause of epidermal division in the late stage of Chinese cabbage leaf stalk macula. Through petiole spot may not pose a direct threat to the overall growth of Chinese cabbage, the cell wall distortion, deformation, and even cell death may heighten vulnerability to other pathogens, such as Ecc, causing significant production issues in Chinese cabbage. Therefore, it is necessary to explore the mechanism of petiole spot, in order to effectively control the occurrence of petiole spot and even other more serious diseases in Chinese cabbage.” (line 282-293)
Secondly, we discuss the role of elements in plant growth and development, especially the disadvantages of elevated N concentrations in line 289-318 as follow.
“ It has been well-documented in several reports that the excessive application of nitrogen fertilizer can exacerbate the occurrence of petiole spot[9]. In contrast to the abiotic stress affecting Chinese cabbage, plant nutrition plays a crucial role in growth and development. Nutrients play a crucial role in plant growth and development, and deficiencies can compromise plant immunity, significantly impacting overall growth [18,19]. For instance, phosphorus (P) enhances photosynthesis, carbohydrate synthesis, and transport. A deficiency in phosphorus results in blocked carbohydrate metabolism, delayed cell division, and hindered cell elongation [20] . Similarly, copper (Cu), an essential nutrient, serves as a crucial component of many plant enzymes and participates in photosynthetic electron transport. Copper deficiency is characterized by slow growth, a lack of green young leaves, and the appearance of dead spots [21]. However, excessive mineral nutrients can disrupt metabolic processes in plant cells, triggering REDOX reactions and causing cell damage. For example, an excess of phosphate fertilizer strengthens plant respiration, consuming significant amounts of sugar and energy, thereby inhibiting growth[20]. Excess Cu binding to S, N, and O ligands inhibits enzyme and functional metalloprotein activity. Additionally, as a REDOX active metal, Cu+ can be oxidized to form highly toxic reactive oxygen species [22-24]. In the same way, Nitrogen, is indispensable for plant growth and reproduction. It serves as a vital component of proteins and nucleic acids and plays a crucial role in carbon fixation during photosynthesis as part of chlorophyll [25]. Plants possess the ability to obtain nitrogen either from the atmosphere as nitrogen (N2) or by absorbing soil nitrogen through their roots. The primary uptake of nitrogen by the vegetable root system occurs from inorganic nitrogen sources, including ammonium and nitrate, as well as small organic nitrogen-containing compounds present in the soil. Research has demonstrated the pivotal role of nitrogen in promoting rapid plant growth and enhancing yield. Consequently, traditional agricultural practices have often relied extensively on the application of excessive nitrogen fertilizers, aiming to maximize vegetable yields [26,27]. However, high nitrogen fertilization rate increased the severity of disease caused by botrytis cinerea in lettuce [28]. In Chinese cabbage, excessive nitrogen can induce petiole spot disease[9]. This dual nature of nutrient effects underscores the delicate balance needed for optimal plant health and productivity. “
Finally, we added the discuss about the relationship between NH4+ and H2O2, which is crucial in the occurrence of petiole spot in line 342-364 as follow.
“A differential gene analysis of cabbage resistant and susceptible to petiole spot under normal and high nitrogen treatments revealed several genes related to plant ROS production that were enriched (Fig 6). Cells with petiole spots on Chinese cabbage exhibit a higher abundance of ROS compared to cells without petiole spots (Fig 7A). ROS are generated in response to a variety of environmental stresses, causing oxidative damage that may lead to cell death [30]. Moreover, an increase in ROS levels was observed in susceptible cabbages upon staining for ROS, and high nitrogen treatments exacerbated this discrepancy (Fig 7B). Notably, NH4+ has been recognized as a catalyst for elevated levels of H2O2 and sequent oxidative stress responses in various plant species, including Arabidopsis[31], the aquatic plant V allisneria natans [32], and numerous other plant species [33]. Interestingly, ROS were concentrated in the intercellular or extracellular regions, mirroring the initial onset site of petiole spot disease. This finding aligns with phenotypic observations (Fig 2). ROS production happens within the apoplast in stress conditions and is accelerated by the activity of NADPH oxidase(NOX) [34].The results of the previous study show that under conditions of high nitrogen concentration, susceptibility traits exhibit heightened cell membrane permeability and plasma membrane peroxidation reactions, as evidenced by increased conductivity and elevated levels of MDA (lipid peroxidation products)[35]. Additionally, the over-application of nitrogen fertilizer can lead to an accumulation of ammonium nitrogen in cabbage petioles, potentially causing damage to the normal cell membrane structure [36]. which causes the release of polyphenols and interact with polyphenol oxidase. This is consistent with previous research results, indicating that Chinese cabbage petiole spot can induce a large number of ROS dependent on excessive accumulation of NH4+.“
In the revised discuss section, we focused more on the relationship of our data with other studies, which were cited as references. The added references are displayed as below.
- Tariqjaveed, M.; Mateen, A.; Wang, S.; Qiu, S.; Zheng, X.; Zhang, J.; Bhadauria, V.; Sun, W. Versatile effectors of phytopathogenic fungi target host immunity. Journal of integrative plant biology 2021, 63, 18.
- Hu, J.; Liu, M.; Zhang, A.; Dai, Y.; Chen, W.; Chen, F.; Wang, W.; Shen, D.; Telebanco-Yanoria, M.J.; Ren, B. Co-evolved plant and blast fungus ascorbate oxidases orchestrate the redox state of host apoplast to modulate rice immunity. Molecular Plant Breeding 2022, 15, 20.
- Carmit, Z.; Zhenzhen, Z.; Gao, Y.G.; Ye, X. Multifunctional Roles of Plant Cuticle During Plant-Pathogen Interactions. Frontiers in Plant Science 2018, 9, 1088-.
- PetraMarschner. Mineral nutrition of higher plants; Mineral nutrition of higher plants: 2013.
- Broadley, M.; Brown, P.; Cakmak, I.; Ma, J.F.; Rengel, Z.; Zhao, F. Beneficial Elements - ScienceDirect. Marschner's Mineral Nutrition of Higher Plants 2012, 72, 249-269.
- Ellsworth, D.S.; Crous, K.Y.; Kauwe, M.G.D.; Verryckt, L.T.; Goll, D.; Zaehle, S.; Bloomfield, K.J.; Ciais, P.; Cernusak, L.A.; Domingues, T.F.; et al. Convergence in phosphorus constraints to photosynthesis in forests around the world. Nature communications 2022, 13, 5005, doi:10.1038/s41467-022-32545-0.
- Mir, A.R.; Pichtel, J.; Hayat, S. Copper: uptake, toxicity and tolerance in plants and management of Cu-contaminated soil. BioMetals 2021.
- Gupta, D.K.; Vandenhove, H.; Inouhe, M. Role of Phytochelatins in Heavy Metal Stress and Detoxification Mechanisms in Plants. Springer Berlin Heidelberg 2013.
- Johnson, A.; Singhal, N.; Hashmatt, M. Metal–Plant Interactions: Toxicity and Tolerance. Spring, 2011.
- Jomova K.; Valko M. Advances in metal-induced oxidative stress and human disease. Toxicology 2011, 283, 65-87.
- Lecompte, F.; Abro, M.A.; Nicot, P.C. Can plant sugars mediate the effect of nitrogen fertilization on lettuce susceptibility to two necrotrophic pathogens: Botrytis cinerea and Sclerotinia sclerotiorum? Plant and Soil;2013, 2013,369(1-2), 387-401.
- Materials and methods More precise information about plant material and experiment variants is required in section 5.1. “One month after transplantation, the seedlings were continuously cultivated with a 10mM NH4NO3 solution for 8 days and the control groups were treated with sterile water.” But how were the plants cultivated this month? How many plants were used in the study, were there any replications, how was the experiment designed? “The samples were named CKA, CKB, TA and TB, respectively.” This is an extremely important point in the article. It must be clearly and unambiguously indicated how the control is denoted, how the N variants are denoted. No information is given about the statistical processing of the data.
Reply:
Thank you for your suggestions. The modifications are as follows: regarding the issue with plant materials, we have provided a detailed description of the cultivation process. Further mention of the naming conventions has also been included in the revised manuscript.
” Two varieties of Chinese cabbage, namely Han Chunwa C8-1 (marked with R) exhibiting high resistance, and the other named Han Chunwa 4-2-3-4 (marked with S) displaying susceptibility, were chosen for the investigation of the Chinese cabbage petiole spot. Both varieties were developed by the Vegetable Institute of Zhejiang Academy of Agricultural Sciences, Hangzhou, China. Seeds of these varieties were sown in 1/2 Murashige and Skoog medium (MS) and placed in a greenhouse with 14 hours of light (25°C) and 10 hours of darkness (16°C) for 15 days. Subsequently, seed-lings were transplanted to floating plates for hydroponic growth over a period of 1 month, utilizing Hoagland's nutrient solution (Farwood Bio, Shanghai). The nutrient solution was then replaced with either ddH2O or a 10 mM NH4NO3 solution for 8 days, three biological replicates per treatment.
Samples collected for transcriptome and proteome analysis were labeled as follows: ddH2O-treated Han Chunwa C8-1 samples: CKA, ddH2O-treated Han Chunwa 4-2-3-4 samples: CKB, 10 mM NH4NO3-treated Han Chunwa C8-1 samples: TA, 10 mM NH4NO3-treated Han Chunwa 4-2-3-4 samples: TB.”
10、References “3. H, C.; W, K.; T, T.; K, R.; K, Z.; H, W.; T, L. Identification of Key Gene Networks Controlling Soluble Sugar and Organic Acid Metabolism During Oriental Melon Fruit Development by Integrated Analysis of Metabolic and Transcriptomic Analyses. Frontiers in plant science 2022, 13, 830517, doi:10.3389/fpls.2022.830517.” Why are the authors written in this way?
Reply:
The insertion of references was carried out using Endnote X8 software, and we did not make further corrections to the inserted references. We apologize for any inconvenience this may have caused. We fix this error:” Cheng, H.; Kong, W.; Tang, T.; Ren, K.; Zhang, K.; Wei, H.; Lin, T. Identification of Key Gene Networks Controlling Soluble Sugar and Organic Acid Metabolism During Oriental Melon Fruit Development by Integrated Analysis of Metabolic and Transcriptomic Analyses. Frontiers in plant science 2022, 13, 830517, doi:10.3389/fpls.2022.830517.” (Line 523-526)
Reviewer 2 Report
Comments and Suggestions for Authors
Comments on the Quality of English LanguageAuthor Response
The second Reviewers' comments:
1: Lines 50-52 - there is no discussion of the current state of knowledge regarding similar research on Chinese cabbage. The authors should supplement the introduction with the latest literature reports in this field
Reply:
Thank you for your suggestions. We have cited the recent literature on petiole blackspot in the introduction. Due to the multifaceted nature of the onset conditions and the prolonged onset period of Chinese cabbage petiole black spot disease, conducting comprehensive studies is challenging. Consequently, there is a scarcity of in-depth research in this field, with most studies primarily relying on phenotypic observations. But we have added the most recent literature on nitrogen to the discussion section.” It has been well-documented in several reports that the excessive application of nitrogen fertilizer can exacerbate the occurrence of petiole spot. In contrast to the abiotic stress affecting Chinese cabbage, plant nutrition plays a crucial role in growth and development. Nutrients play a crucial role in plant growth and development, and deficiencies can compromise plant immunity, significantly impacting overall growth. For instance, phosphorus (P) enhances photosynthesis, carbohydrate synthesis, and transport. A deficiency in phosphorus results in blocked carbohydrate metabolism, delayed cell division, and hindered cell elongation. Similarly, copper (Cu), an essential nutrient, serves as a crucial component of many plant enzymes and participates in photosynthetic electron transport. Copper deficiency is characterized by slow growth, a lack of green young leaves, and the appearance of dead spot. However, excessive mineral nutrients can disrupt metabolic processes in plant cells, triggering REDOX reactions and causing cell damage. For example, an excess of phosphate fertilizer strengthens plant respiration, consuming significant amounts of sugar and energy, thereby inhibiting growth. Excess Cu binding to S, N, and O ligands inhibits enzyme and functional metalloprotein activity. Additionally, as a REDOX active metal, Cu+ can be oxidized to form highly toxic reactive oxygen species. In the same way, Nitrogen, is indispensable for plant growth and reproduction. It serves as a vital component of proteins and nucleic acids and plays a crucial role in carbon fixation during photosynthesis as part of chlorophyll. Plants possess the ability to obtain nitrogen either from the atmosphere as nitrogen (N2) or by absorbing soil nitrogen through their roots. The primary uptake of nitrogen by the vegetable root system occurs from inorganic nitrogen sources, including ammonium and nitrate, as well as small organic nitrogen-containing compounds present in the soil. Research has demonstrated the pivotal role of nitrogen in promoting rapid plant growth and enhancing yield. Consequently, traditional agricultural practices have often relied extensively on the application of excessive nitrogen fertilizers, aiming to maximize vegetable yields. However, high nitrogen fertilization rate increased the severity of disease caused by botrytis cinerea in lettuce. In Chinese cabbage, excessive nitrogen can induce petiole spot disease. This dual nature of nutrient effects underscores the delicate balance needed for optimal plant health and productivity.” (Line 289-318).
2: Lines 64-77 - there is no research hypothesis and a precisely defined research goal. The description presented in this paragraph is, in my opinion, a fragment of an abstract or summary, but does not indicate what type of research problem was undertaken. Please complete the final part of the introduction with the research hypothesis and the aim of the work.
Reply:
Thank you for your suggestions. We also recognize this issue, and accordingly, based on your advice, we have completed the research hypotheses and objectives in the introduction section.
“It is noteworthy that previous research on Chinese cabbage petiole spot mainly focuses on field investigations and physiological experiments and has clarified the influencing factors influencing factors contributing to the onset of the disease. However, the precise mechanism underlying Chinese cabbage petiole spot, remains unclear, which needs more experimental evidence to confirm. Although the promotion of Chinese cabbage petiole spot by excessive nitrogen fertilizer is well-established, the precise mechanism remains elusive.
” To unravel the intricate mechanisms of Chinese cabbage petiole spot, we meticlously examine the morphology of petiole spot in resistant and susceptible varieties to petiole spot. we Transcriptome and proteome were employed to identify differentially expressed genes (DEGs) and differentially expressed proteins (DEPs) among resistant and susceptible varieties upon high nitrogen treatment. The findings of our study could provide new insights into the breeding strategies for resistance to petiole spot in Chinese cabbage.” (line 71-76)
3: Lines 85 - please remove the space from the bracket
Reply:
Thank you for your suggestions, we revised it accordingly.
Reviewer #4: Line 130 - grammar - a sentence in the passive voice would sound better.
Reply:
Thank you for your suggestions, we revised it accordingly.
5: Line 256-260 - there is no reference of the obtained results to other scientific works.
Reply:
Thank you for your suggestions, we revised it accordingly.
6: Figure 1 - there is a different designation in the text (methodology) and a different one on the Figure 1. Please standardize.
Reply:
Thank you for your suggestions. We have corrected the names in the article.
“Two varieties of Chinese cabbage, namely Han Chunwa C8-1 (marked with R) exhibiting high resistance, and the other named Han Chunwa 4-2-3-4 (marked with S) displaying susceptibility, were chosen for the investigation of the Chinese cabbage petiole spot. Both varieties were developed by the Vegetable Institute of Zhejiang Academy of Agricultural Sciences, Hangzhou, China. Seeds of these varieties were sown in 1/2 Murashige and Skoog medium (MS) and placed in a greenhouse with 14 hours of light (25°C) and 10 hours of darkness (16°C) for 15 days. Subsequently, seed-lings were transplanted to floating plates for hydroponic growth over a period of 1 month, utilizing Hoagland's nutrient solution (Farwood Bio, Shanghai). The nutrient solution was then replaced with either ddH2O or a 10 mM NH4NO3 solution for 8 days, three biological replicates per treatment.
Samples collected for transcriptome and proteome analysis were labeled as follows: ddH2O-treated Han Chunwa C8-1 samples: CKA, ddH2O-treated Han Chunwa 4-2-3-4 samples: CKB, 10 mM NH4NO3-treated Han Chunwa C8-1 samples: TA, 10 mM NH4NO3-treated Han Chunwa 4-2-3-4 samples: TB” (line 381-395)
Which part of the Chinese cabbage was chosen for testing? Outer leaves or middle leaves? Do the authors know in which area of Chinese cabbage changes occur the earliest? Are these just the outer leaves? Please explain when discussing the results presented in Figure 1.
We conducted tests using the outer leaves of cabbage. It is important to note that petiole black spot occurs exclusively on the petiole and is not observed on the soft part of the leaf. Regarding the other issues you mentioned, we have made modifications in the experimental methods and the discussion section.
“Interestingly, according to our observation in the field, no petiole spots were observed on the rosette leaves of Chinese cabbage. However, the number of petiole spots gradually increased from the outermost to the inner leaf layers of the head (data not shown).” (Line 263-266).
7: Line 341 - please put a dot instead of a comma after the brackets.
Reply:
As suggested, we revised it accordingly.
8: Line 344 - please explain the abbreviations
Reply:
We revised it accordingly. And cited the reference provided by the reviewer.
9: Line 350 - grammar - a sentence in the passive voice would sound better.
Reply:
As suggested, we revised it accordingly.
10: Line 389-392 - please complete the description of the experiments performed (NH4+, GS, GDS) or provide literature references.
Reply:
Thank you for your suggestions. We have incorporated the relevant experimental descriptions into the experimental methods section.
“5.5. Determination of ammonium nitrogen content, glutamine synthetase and glutamate dehydrogenase activities in plants
The content of ammonium nitrogen (NH4+) in plants was determined using a Plant Ammonium Nitrogen Content Assay Kit (Boxbio, Beijing) according to the manufacturer’s instructions. Weigh 0.1 g of Chinese cabbage petiole and combine it with 1 mL of the extraction solution. Homogenize the mixture at room temperature, then centrifuge at 10,000 g at 4℃ for 10 minutes, using the supernatant as the test sample. Preheat the spectrophotometer for at least 30 minutes, set the wavelength to 630 nm, and zero with distilled water. In a 1.5 mL centrifuge tube, add 20 μL of the sample or extract (blank control), along with 490 μL of Reagent I and 490 μL of Reagent II. Thoroughly mix the components and allow the color reaction to occur at 25°C for 20 minutes. Transfer 1 mL of the reaction solution to a 1 mL glass cuvette. Measure the absorbance at 630 nm, record the value as A, and calculate ∆A = A (measurement) - A(blank).
Glutamine synthetase (GS) activities were measured according to the instructions from Solarbio. Weigh 0.1g of Chinese cabbage petiole and add 1mL of the extraction solution. Homogenize the mixture in an ice bath and centrifuge at 8000g for 10 minutes at 4℃. Collect the supernatant and keep it on ice for subsequent measurements. Preheat the zymograph for 30 minutes, adjusting the wavelength to 540nm and zeroing it with distilled water. In the measurement tube, combine 160 μL of Reagent I, 70 μL of Reagent III, and 70 μL of the sample. For the control tube, mix 160 μL of Reagent II, 70 μL of Reagent III, and 70 μL of the sample. After thorough mixing and incubating in a 25℃water bath, add 100 μL of Reagent 4 to both the measurement and control tubes. Allow them to stand for 10 minutes, then centrifuge the samples at 5000 g for 10 minutes at room temperature. Transfer 200 μL of supernatant from the sample into a 96-well UV plate and determine the absorbance at 540 nm, recording it as B. Utilize the formula ΔB = B (measurement) - B (control) to calculate the difference in absorbance between the assay and control samples.
glutamate dehydrogenase (GDH) activities were also measured according to the instructions from Solarbio. Weigh approximately 0.1g of tissue and combine it with 1mL of extraction solution for ice bath homogenization. Centrifuge at 8000g and 4℃ for 10 minutes, then carefully collect the supernatant, keeping it on ice for subsequent measurements. Preheat the zymograph for at least 30 minutes, adjusting the wavelength to 340 nm and zeroing it with distilled water. In a 96-well UV plate, add 10 μL of the sample and 190 μL of the working solution. Ensure thorough mixing and immediately record the absorbance value C1 at 340 nm for 20 s, followed by the absorbance value C2 after 5 minutes and 20 seconds. Calculate ΔC = C1 - C2.” (line 443-475).